# REINFORCING SPATIO-TEMPORAL GRAPH NEURAL NETWORKS WITH A PHYSICS REWARD ORACLE

## ABSTRACT

Modeling and predicting spatio-temporal dynamical systems are pivotal tasks in science and engineering that pose significant challenges. Graph Neural Networks (GNNs) have emerged as a mainstream approach for this purpose, renowned for their effectiveness in capturing complex spatial dependencies. However, these models often exhibit insufficient robustness and unreliable uncertainty estimation when confronted with out-of-distribution (OOD), unseen, or perturbed scenarios, limiting their fidelity in critical applications. To address this limitation, this paper proposes a novel training framework `PhyRL`, Physics-Guided Reinforcement Learning, designed to fundamentally enhance the OOD generalization of predictive models. At the core of our approach is an automated "Physics Reward Oracle," which leverages physical priors to provide verifiable, quantitative rewards for multiple candidate future trajectories generated by the model. The reward function holistically evaluates each trajectory based on its physical consistency, robustness against perturbations, and the reliability of its uncertainty estimates. During training, we leverage this reward signal within a general reinforcement learning paradigm to directly optimize the model. This approach compels the model to move beyond mere data fitting, encouraging it to learn and internalize the intrinsic physical properties and robust behaviors of the underlying system. Experiments demonstrate that our framework significantly improves the OOD generalization, prediction accuracy, and quality of uncertainty quantification for state-of-the-art spatio-temporal GNN architectures in complex prediction tasks. This research offers a new perspective on tackling fundamental challenges in scientific computing, particularly in enhancing the physical fidelity and robustness of graph-structured models. Our codes are available at https://anonymous.4open.science/r/PhyRL-2525/

## 1 INTRODUCTION

Modeling and predicting spatio-temporal dynamical systems (Wu et al., 2023b; Shi et al., 2015; Yu et al., 2018; Chen et al., 2022) are pivotal tasks in numerous domains of science and engineering, from long-term climate evolution (Bi et al., 2023; Wu et al., 2025; Yuval & O'Gorman, 2020) in climate science and aircraft design in computational fluid dynamics, to multi-agent trajectory planning in autonomous driving. Traditionally, these systems are described by complex partial differential equations (PDEs) (Chen & Shaw, 2001; Krantz, 2018) and solved using computationally expensive numerical simulation methods. In recent years, data-driven approaches have emerged as a promising alternative. Among these, deep learning, and specifically Graph Neural Networks (GNNs) (Fan et al., 2019; Pfaff et al., 2020), provides a powerful new paradigm for efficiently learning complex evolutionary patterns from data.

Consequently, models based on Graph Neural Networks have become a mainstream approach for simulating physical systems (Pfaff et al., 2020; Gao et al., 2025; Lam et al., 2023; Poli et al., 2019). The unique strength of GNNs lies in their ability to naturally handle non-Euclidean data defined by meshes (Guskov et al., 2002; Pfaff et al., 2020), particles, or arbitrary graph structures, and to explicitly model the spatial dependencies and interactions among system components. Thanks to these powerful inductive biases, spatio-temporal GNNs demonstrate exceptional performance on various benchmarks, enabling accurate short-term predictions while accelerating simulations by several orders of magnitude (Yu et al., 2018; Mohan et al., 2020; Yu et al., 2018).

However, despite these successes, current GNN models face fundamental challenges when applied to critical real-world tasks. First, they *lack robustness*, being highly sensitive to minor perturbations in the input data, which can lead to catastrophic, physically implausible deviations in predicted trajectories. Second, their *out-of-distribution (OOD) generalization is weak* (Göring et al., 2024; Rame et al., 2022; Wu et al., 2024a;b). These models are essentially powerful curve-fitters that learn statistical correlations within the training distribution rather than the underlying, universal physical laws. Their performance degrades sharply when they encounter new scenarios that differ from the training distribution. Finally, the *uncertainty estimates* (Li et al., 2022; Wang et al., 2022;?; Liu et al., 2021) they provide are often *unreliable*; when facing unknown situations, they not only produce incorrect predictions but also frequently exhibit overconfidence, which is unacceptable in safety-critical domains where risk assessment is paramount.

To address these limitations, this paper proposes a novel training framework: *Physics-Guided Reinforcement Learning* (`PhyRL`). The core idea is to move beyond conventional loss functions based on pixel-wise or numerical errors (e.g., Mean Squared Error) and instead leverage reinforcement learning to directly shape the model's *behavior*. To this end, we design and implement an automated *Physics Reward Oracle*. This oracle translates abstract physical principles into a verifiable, quantitative reward signal used to holistically evaluate every candidate trajectory generated by the model. Its evaluation criteria directly address the aforementioned challenges: (1) *physical consistency* to improve OOD generalization; (2) *robustness against perturbations* to enhance model stability; and (3) *reliability of uncertainty* to ensure the model's *honesty* in uncharted domains. This approach aims to drive the model from passive data fitting toward actively learning and *internalizing* the intrinsic physical properties and robust behaviors of the system.

The main contributions of this work are summarized as follows:

- We propose `PhyRL`, a novel training framework that, for the first time, integrates physics-guided reinforcement learning with spatio-temporal GNNs to fundamentally address core deficiencies in robustness and OOD generalization.

- We design and implement a general-purpose Physics Reward Oracle, the core engine of `PhyRL`, that translates high-level physical concepts (e.g., consistency, robustness) into concrete, computable reward functions to guide model training.

- Through extensive experiments on several complex spatio-temporal dynamics simulation tasks, we demonstrate that `PhyRL` significantly improves the prediction accuracy, OOD generalization, and quality of uncertainty quantification of GNN models compared to state-of-the-art methods.

## 2 METHOD

### 2.1 PROBLEM FORMULATION

In this study, we represent the state of a spatio-temporal dynamical system at discrete time points as a graph. Specifically, at a time step $t$, the instantaneous state of the system is described by a graph $G_t = (\mathcal{V}, \mathcal{E}, \mathbf{X}_t)$. Here, $\mathcal{V} = \{v_1, \ldots, v_N\}$ is a set of $N$ nodes representing the discrete units within the physical system, such as particles, grid points, or sensors. The set $\mathcal{E} \subseteq \mathcal{V} \times \mathcal{V}$ contains the edges, which encode the spatial proximity or physical interactions between these units. The feature matrix $\mathbf{X}_t \in \mathbb{R}^{N \times d}$ stores the physical states, where its $i$-th row vector, $\mathbf{x}_{i,t} \in \mathbb{R}^d$, represents the $d$-dimensional attributes of node $v_i$ at time $t$, such as position, velocity, or temperature.

The central task of our work is to learn a deep learning model capable of accurately predicting the future evolution of this system. Formally, we aim to learn a GNN model, parameterized by $\theta$, denoted as $f_\theta$. This model takes a sequence of the system's state graphs over the past $k$ consecutive time steps, $\{G_{t-k+1}, \ldots, G_t\}$, as input and autoregressively predicts the sequence of state graphs for the next $T$ future time steps. Let $\hat{\mathcal{G}}_{t+1:t+T} = \{\hat{G}_{t+1}, \ldots, \hat{G}_{t+T}\}$ denote the predicted future trajectory, where each predicted state graph $\hat{G}_\tau$ shares the same graph topology $(\mathcal{V}, \mathcal{E})$ but features a new node attribute matrix $\hat{\mathbf{X}}_\tau$. The entire prediction process is thus formulated as:

$$\hat{\mathcal{G}}_{t+1:t+T} = f_\theta(\{G_{t-k+1}, \ldots, G_t\}). \tag{1}$$

The conventional supervised learning paradigm optimizes the model parameters $\theta$ by minimizing the cumulative error between the predicted trajectory and the ground-truth trajectory $\hat{\mathcal{G}}^*_{t+1:t+T}$. The

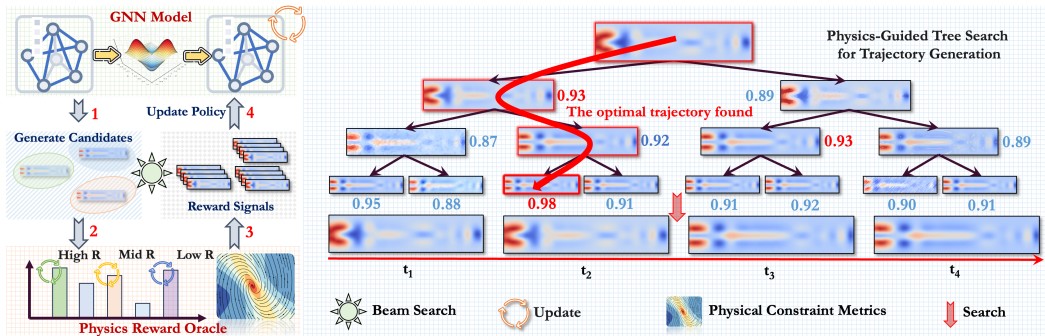

Figure 1: Conceptual illustration of our proposed framework, which encompasses two distinct processes. **(Left)** The physics-guided training loop, where a GNN policy is optimized via reinforcement learning. The model learns by generating candidate trajectories (①), which are evaluated by a Physics Reward Oracle (②) to produce reward signals (③) that guide the policy update (④). **(Right)** The inference process, where the trained GNN is combined with a Physics-Guided Tree Search algorithm to generate the optimal future trajectory. The search explores possible futures, using the reward scores from the oracle at each step to identify the most physically plausible path.

objective function is typically the Mean Squared Error (MSE):

$$\theta^* = \arg\min_\theta \mathbb{E}_{\mathcal{G}} \left[ \sum_{\tau=t+1}^{t+T} \sum_{i=1}^{N} \left\| \hat{\mathbf{x}}_{i,\tau} - \mathbf{x}_{i,\tau}^* \right\|_2^2 \right]. \tag{2}$$

However, as previously discussed, an objective function that relies solely on data fitting often leads to poor generalization and robustness when the model is confronted with out-of-distribution (OOD) scenarios. The goal of this paper is therefore to design a new training paradigm that moves beyond this point-wise error minimization, instead employing reinforcement learning to guide the model $f_\theta$ toward generating trajectories that are more physically plausible and robust.

## 2.2 THE PHYSICS-GUIDED REINFORCEMENT LEARNING FRAMEWORK

To overcome the inherent limitations of the traditional supervised learning paradigm (Eq. 2), we reformulate the trajectory prediction problem as a dual-process framework, encompassing both a training phase to learn the physical dynamics and an inference phase to generate optimal predictions. This entire framework is illustrated in Figure 1.

**Training via Reinforcement Learning (Figure 1, Left)** In the training phase, we construct a closed-loop reinforcement learning environment. The GNN model $f_\theta$ serves as the **policy** $\pi_\theta$, which acts as an agent. The policy's task is to generate a complete future trajectory, $\tau$, which constitutes its **action**. This action is taken based on the current **state**, which is the sequence of past observations. The quality of the generated trajectory is then evaluated by a scalar **reward** signal, $R(\tau)$, provided by our **Physics Reward Oracle**. As shown in the figure, the training cycle proceeds in four steps: (①) the policy generates a batch of diverse candidate trajectories; (②) the Physics Reward Oracle evaluates each trajectory based on physical priors and assigns a reward score; (③) these scores are collected as reward signals; and (④) a policy gradient algorithm, such as PPO, uses these signals to update the policy parameters $\theta$. This process iteratively reinforces the model's ability to generate trajectories that are physically plausible.

**Inference via Physics-Guided Tree Search (Figure 1, Right)** Once the GNN policy is trained, the objective at inference time is to generate the single most physically consistent future trajectory. A simple autoregressive rollout can be brittle and accumulate errors. Therefore, we employ a **Physics-Guided Tree Search** algorithm. This process begins with an initial state (the root of the tree) and explores possible future states over a time horizon $(t_1, t_2, \dots)$. At each step, the trained GNN proposes several potential next states, creating branches in the search tree. The key insight of our framework is the **reuse of the Physics Reward Oracle during this search**. The oracle provides an immediate reward score for each potential state transition, allowing the search algorithm (e.g., beam search) to intelligently prune the search space. It preferentially expands paths with the highest

cumulative reward scores, effectively navigating toward the most physically plausible future. The final output is the single trajectory with the highest overall score, identified by backtracking along the optimal path found by the search (visualized as the red serpentine line).

**Synergy of Training and Inference** The two processes work in synergy. The RL training loop teaches the GNN to act as an effective proposal generator, imbuing it with an implicit understanding of the physical "value landscape." The tree search at inference then leverages this powerful learned prior, using the explicit guidance of the reward oracle to efficiently search this landscape and pinpoint the optimal trajectory. This combination ensures that our model not only learns the underlying physical laws but also effectively utilizes them to make robust and coherent predictions.

### 2.3 THE PHYSICS REWARD ORACLE

The Physics Reward Oracle serves as the cornerstone of our framework, acting as the bridge between the generative capabilities of the GNN policy and the optimization objective of the reinforcement learning algorithm. Its function is to map any given candidate trajectory $\tau$ to a scalar reward $R(\tau)$ that quantifies its physical fidelity. This reward is not a monolithic metric but rather a composite function, designed to be both comprehensive and configurable. We formulate the total reward as a weighted combination of three distinct components, each targeting a critical aspect of physical plausibility:

$$R(\tau) = w_c \cdot R_{\text{consistency}}(\tau) + w_r \cdot R_{\text{robustness}}(\tau) + w_u \cdot R_{\text{uncertainty}}(\tau), \tag{3}$$

where $w_c, w_r, w_u$ are scalar weights that balance the relative importance of each physical desideratum.

The first and most critical component, the **consistency reward** $R_{\text{consistency}}$, measures the degree to which a trajectory adheres to the underlying physical laws of the system. The formulation of this reward is context-dependent, adapting to the available physical priors. For systems governed by known partial differential equations (PDEs), such as the advection-diffusion dynamics common in fluid mechanics, we quantify consistency by the PDE residual. Given a predicted state evolution $\hat{\mathbf{X}}(v, t')$ for nodes $v \in \mathcal{V}$ over time $t' \in [t, t + T]$, the PDE residual operator $\mathcal{R}_{\text{PDE}}$ is defined as the extent to which the governing equation is violated. For an advection-diffusion process, this is:

$$\mathcal{R}_{\text{PDE}}(\hat{\mathbf{X}}) = \frac{\partial \hat{\mathbf{X}}}{\partial t'} + (\mathbf{w} \cdot \nabla)\hat{\mathbf{X}} - D\nabla^2 \hat{\mathbf{X}}. \tag{4}$$

We compute a consistency loss by integrating the squared norm of this residual over the trajectory's spatio-temporal domain, where differential operators are approximated using numerical schemes on the discrete graph structure. In scenarios where explicit PDEs are unavailable but fundamental symmetries are known, such as in N-body systems or molecular dynamics, we leverage the principle of **equivariance**. Given a transformation group $\mathcal{T}$ (e.g., the Euclidean group SE(3) for rotations and translations), a perfectly equivariant model $f_\theta$ must satisfy $T(f_\theta(G)) = f_\theta(T(G))$ for any $T \in \mathcal{T}$. We quantify the violation of this property by computing an equivariance error over the trajectory. Irrespective of the source, we transform the non-negative error $\mathcal{L}_{\text{consistency}}$ into a bounded reward using an exponential kernel:

$$R_{\text{consistency}}(\tau) = \exp(-\lambda_c \cdot \mathcal{L}_{\text{consistency}}(\tau)), \tag{5}$$

where $\lambda_c$ is a positive scaling hyperparameter.

Beyond consistency, a desirable property of any physical model is **robustness** against minor perturbations. The reward component $R_{\text{robustness}}$ is designed to encourage this stability. We quantify robustness by measuring the sensitivity of the model's output to small, stochastic perturbations in its input. Given an initial state sequence ending in $G_t$ with features $\mathbf{X}_t$, we generate a perturbed counterpart $G_t'$ by adding zero-mean Gaussian noise $\delta \sim \mathcal{N}(0, \sigma^2 I)$ to $\mathbf{X}_t$. The model then produces two trajectories, $\tau = f_\theta(G_t)$ and $\tau' = f_\theta(G_t')$. The robustness loss is defined as the normalized squared deviation between these trajectories, and the corresponding reward is:

$$R_{\text{robustness}}(\tau) = \exp\left(-\lambda_r \cdot \frac{\|\tau - \tau'\|_F^2}{\|\delta\|_F^2}\right), \tag{6}$$

where $\|\cdot\|_F$ denotes the Frobenius norm and $\lambda_r$ is a scaling factor. This reward incentivizes the model to learn smooth functions that are less susceptible to noise, a hallmark of well-behaved physical systems.

Finally, to ensure model fidelity in unknown scenarios, the **uncertainty reward** $R_{\text{uncertainty}}$ encourages the model to be "honest" about its own predictive confidence. We estimate the model's epistemic uncertainty using Monte-Carlo Dropout, performing $K$ stochastic forward passes to generate an ensemble of trajectories $\{\tau_k\}_{k=1}^K$. The uncertainty $\mathcal{U}(\tau)$ is quantified as the variance of this ensemble, for instance, the trace of the trajectory covariance matrix. This reward is selectively applied to incentivize high uncertainty exclusively for out-of-distribution (OOD) inputs. We formalize this as:

$$R_{\text{uncertainty}}(\tau) = \mathbb{I}[G_t \in \mathcal{D}_{\text{OOD}}] \cdot \mathcal{U}(\tau), \tag{7}$$

where $\mathbb{I}[\cdot]$ is the indicator function and $\mathcal{D}_{\text{OOD}}$ represents a set of pre-defined OOD samples. This component discourages the model from making overconfident yet incorrect predictions when faced with novel physical regimes, a critical feature for trustworthy scientific modeling.

### 2.4 Model Optimization via Reinforcement Learning

Given the scalar reward signal $R(\tau)$ from the Physics Reward Oracle, we require a mechanism to translate this signal into an optimizable objective for updating the GNN policy $\pi_\theta$. To this end, we employ **Proximal Policy Optimization (PPO)**, a state-of-the-art policy gradient algorithm renowned for its sample efficiency and stable training dynamics.

The fundamental objective of our reinforcement learning formulation is to find the policy parameters $\theta$ that maximize the expected reward:

$$J(\theta) = \mathbb{E}_{\tau \sim \pi_\theta}[R(\tau)]. \tag{8}$$

Directly optimizing Eq. 8 with vanilla policy gradients is often unstable due to the high variance of the gradient estimators. To mitigate this, we introduce an **advantage function**, $A^{\pi_\theta}(s_t, \tau)$, which quantifies how much better a given trajectory $\tau$ is compared to the average expected reward from state $s_t$:

$$A^{\pi_\theta}(s_t, \tau) = R(\tau) - V^{\pi_\theta}(s_t), \tag{9}$$

where $V^{\pi_\theta}(s_t)$ is the state-value function. In our macro-action setting, we estimate this value function using a simple yet effective baseline $b(s_t)$, calculated as the mean reward over a batch of $M$ candidate trajectories generated from the same state $s_t$. The resulting advantage estimate for a trajectory $\tau_i$ is thus:

$$\hat{A}(\tau_i) = R(\tau_i) - \left( \frac{1}{M} \sum_{j=1}^{M} R(\tau_j) \right). \tag{10}$$

PPO further enhances training stability by constraining the magnitude of policy updates at each step. This is achieved through a clipped surrogate objective function that depends on the probability ratio between the new policy $\pi_\theta$ and the old policy $\pi_{\theta_{\text{old}}}$ used to generate the data:

$$r_t(\theta) = \frac{\pi_\theta(\tau|s_t)}{\pi_{\theta_{\text{old}}}(\tau|s_t)}. \tag{11}$$

The final policy loss, which we aim to minimize, is then formulated as the negative of the PPO surrogate objective:

$$L^{\text{CLIP}}(\theta) = -\hat{\mathbb{E}}_t \left[ \min \left( r_t(\theta)\hat{A}_t, \text{clip}(r_t(\theta), 1 - \epsilon, 1 + \epsilon)\hat{A}_t \right) \right], \tag{12}$$

where the clip function constrains the ratio $r_t(\theta)$ to the interval $[1 - \epsilon, 1 + \epsilon]$, with $\epsilon$ being a small hyperparameter. This objective provides a pessimistic lower bound on the policy improvement, so discouraging destructively large updates and ensuring a more monotonic and stable learning process.

## 3 Experiments

### 3.1 Experimental Setup

To conduct a rigorous and comprehensive evaluation of our proposed framework, we designed experiments spanning three categories of tasks with distinct physical priors.

### 3.1.1 N-BODY SYSTEMS: DYNAMICS FROM SYMMETRY PRIORS

To assess our framework's ability to handle systems governed by symmetries rather than explicit PDEs, we employ the classic N-body gravitational simulation task. The **dataset** setup follows Satorras et al. (2021) (Satorras et al., 2021), simulating the motion of $N = 5$ charged particles under Coulomb forces. The out-of-distribution (OOD) test set is constructed by varying the number of particles in the system. Following the benchmark established by Xu et al. (Xu et al., 2024) for related tasks, we compare our method (`PhyRL`) against a suite of state-of-the-art equivariant GNN architectures, including: *Linear* (Satorras et al., 2021), *SE(3)-Transformer* (Fuchs et al., 2020), *TFN* (Thomas et al., 2018), *MPNN* (Gilmer et al., 2017), *RF* (Köhler et al., 2019), *ClofNet* (Du et al., 2022), and our direct backbone competitor, *EGNN* (Satorras et al., 2021).

### 3.1.2 PDE-GOVERNED SYSTEMS: FLUID DYNAMICS AND REAL-WORLD APPROXIMATIONS

To test the framework's performance on systems described by partial differential equations, we select two benchmarks: a simulated fluid governed by the exact Navier-Stokes equations, and real-world sea temperature data whose dynamics can be approximated by the advection-diffusion equation.

❶ **Navier-Stokes Dataset:** We use a 2D vorticity simulation. The OOD scenario is created using a higher viscosity coefficient unseen during training. (Li et al., 2021)

❷ **Sea Surface Temperature (SST) Dataset:** We adopt a real-world satellite observation dataset, with a setup inspired by de Bézenac et al. (De Bézenac et al., 2019). OOD generalization is tested across geographical regions (training on the Atlantic, testing on the Pacific).

For these two tasks, we adopt the strong set of spatio-temporal prediction baselines used by Xing et al. in the HelmFluid (Xing et al., 2023). This includes the numerical method *DARTS* (Ruzanski et al., 2011), the general vision backbone *U-Net*, and a series of advanced neural operators and forecasting models such as *FNO* (Li et al., 2021), *MWT* (Gupta et al., 2021), *U-NO* (Ashiqur Rahman et al., 2022), and *LSM* (Wu et al., 2023a). For the SST task, we additionally include the *Physics-Inspired CNN* (De Bézenac et al., 2019) as a strong, domain-specific baseline.

### 3.1.3 IMPLEMENTATION DETAILS AND EVALUATION METRICS

To ensure a fair comparison, all GNN-based methods, including our own, share the same EGNN backbone architecture. All models are trained to convergence using the Adam optimizer. Model performance is assessed across multiple dimensions: **prediction accuracy** is quantified by the Normalized Root Mean Squared Error (NRMSE); **robustness** is evaluated by the error increase after adding noise of varying intensities to the initial state; and **uncertainty quality** is measured by the Expected Calibration Error (ECE). Furthermore, we conduct ablation studies on the key reward components of our method (`PhyRL (w/o $R_c$)` and `PhyRL (w/o $R_r$)`).

## 3.2 MAIN RESULTS

We evaluated our proposed framework on all three benchmark tasks against a range of strong baselines, with the quantitative results presented in Table 2 and Table 1. For the N-body system governed by symmetries (Table 2), our method slightly outperforms the best-performing baseline, EGNN (MSE), on the in-distribution (ID) test with an NRMSE of 0.0195 versus 0.0213. The true advantage of our approach, however, becomes evident in the more challenging out-of-distribution (OOD) scenario. When the number of particles is altered, the performance of the purely supervised EGNN model degrades sharply, with its error increasing to 0.0986. In stark contrast, our physics-guided reinforcement learning framework demonstrates superior generalization, maintaining the OOD error at an impressive 0.0314, which constitutes a relative error reduction of over 68%. This result provides strong evidence that by rewarding physical consistency (equivariance in this case), our model learns not just patterns in the data, but the underlying, generalizable laws of the system's dynamics.

Our method exhibits consistent superiority on PDE-governed systems as well (Table 1). Across both the simulated Navier-Stokes and the real-world SST datasets, our model achieves the lowest prediction error in all ID and OOD scenarios. Notably, in the OOD tests, the performance of conventional data-driven methods like GNN (MSE) and U-Net deteriorates significantly. While the PINN-style approach improves generalization to some extent by incorporating a PDE soft constraint

Table 1: Quantitative results on PDE-governed systems. We report Normalized Root Mean Squared Error (NRMSE) on the simulated Navier-Stokes dataset and the real-world Sea Surface Temperature (SST) dataset. The OOD scenarios involve unseen viscosity (Navier-Stokes) and unseen geography (SST). Our method consistently outperforms strong baselines from both numerical methods and deep learning, highlighting its versatility and robustness. The best results are in **bold**; the second-best are underlined.

| | Navier-Stokes | | Sea Surface Temperature (SST) | |
|---|---|---|---|---|
| **Methods** | NRMSE (ID, ↓) | NRMSE (OOD, ↓) | NRMSE (ID, ↓) | NRMSE (OOD, ↓) |
| DARTS (Numerical) | 0.1582 | 0.2845 | 0.2134 | 0.3541 |
| U-Net | 0.0815 | 0.2511 | 0.1152 | 0.2988 |
| FNO | 0.0754 | 0.1982 | 0.1088 | 0.2412 |
| MWT | 0.0781 | 0.2015 | 0.1101 | 0.2503 |
| U-NO | 0.0722 | 0.1854 | 0.1053 | 0.2355 |
| LSM | 0.0695 | 0.1801 | 0.1094 | 0.2489 |
| Physics-Inspired CNN | — | — | 0.0985 | 0.1892 |
| GNN (MSE) | 0.0701 | 0.2243 | 0.1002 | 0.2764 |
| PINN-style | 0.0734 | 0.1755 | 0.1031 | 0.2105 |
| PhyRL | **0.0611** | **0.0894** | **0.0913** | **0.1422** |
| PhyRL (w/o $R_c$) | 0.0715 | 0.1723 | 0.1011 | 0.2088 |

(e.g., reducing the OOD error from 0.2243 to 0.1755 on Navier-Stokes), our method further slashes this error to 0.0894 through the reinforcement learning paradigm. This suggests that using physical priors as a reward signal to guide behavior is a more effective and robust mechanism for knowledge injection than simply adding them as a loss term. Furthermore, our ablation study (**PhyRL (w/o $R_c$)**) corroborates this finding: without the consistency reward, the model's OOD performance regresses to a level comparable with the PINN-style baseline, confirming the central role of $R_{\text{consistency}}$ in achieving superior generalization.

### 3.2.1   ROBUSTNESS ANALYSIS

Beyond accuracy on clean data, a critical characteristic of a physical model is its stability in the presence of input perturbations. To quantify the effectiveness of our framework in enhancing model robustness, we conduct a systematic perturbation analysis. Specifically, we add Gaussian noise of varying intensity ($\sigma$) to the initial states of test samples and evaluate the degradation in prediction error for each model. The results are visualized in Figure 2.

The plots clearly show that while all models inevitably experience performance degradation as input noise increases, their rates of decay differ significantly. Across all three datasets, the **GNN (MSE)** baseline, trained solely on mean squared error, exhibits the highest sensitivity to noise, as indicated by the steepest error curve. This suggests that the learned dynamics mapping is brittle and unstable. In stark contrast, our

Table 2: Quantitative results on the N-body simulation task. We report Normalized Root Mean Squared Error (NRMSE) for both in-distribution (ID) and out-of-distribution (OOD) scenarios. Our method significantly outperforms all state-of-the-art equivariant baselines, especially in the challenging OOD setting. The best results are in **bold**; the second-best are underlined.

| | N-Body System | |
|---|---|---|
| **Methods** | NRMSE (ID, ↓) | NRMSE (OOD, ↓) |
| Linear | 0.8712 | 1.2543 |
| MPNN | 0.0954 | 0.1872 |
| TFN | 0.0811 | 0.1655 |
| SE(3)-Tr | 0.0652 | 0.1421 |
| RF | 0.0588 | 0.1309 |
| ClofNet | 0.0415 | 0.1152 |
| EGNN (MSE) | 0.0213 | 0.0986 |
| PhyRL | **0.0195** | **0.0314** |
| PhyRL (w/o $R_c$) | 0.0221 | 0.0955 |

method (**PhyRL**) demonstrates exceptional robustness across all noise levels, with its performance curve remaining the flattest among all competitors. This provides strong evidence that by incorporat-

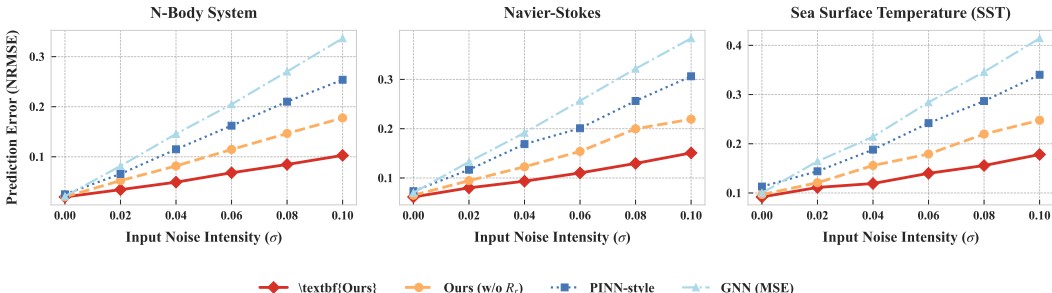

Figure 2: Robustness analysis with respect to input noise. We add Gaussian noise of varying intensity ($\sigma$) to the initial state and report the Normalized Root Mean Squared Error (NRMSE). Across all three datasets, our method (**Ours**) demonstrates significantly higher resilience to perturbations compared to all baselines. The flatter slope of our model's error curve indicates superior stability, a direct result of the robustness reward component in our training framework.

ing the $R_{\text{robustness}}$ reward term, our framework successfully guides the model to learn a smoother and more resilient dynamics function. It is also noteworthy that the performance of our ablation variant (`PhyRL (w/o` $R_r$`)`) consistently lies between that of the purely supervised baseline and our full model, directly validating that the robustness reward is the key factor in achieving this superior stability.

### 3.2.2 UNCERTAINTY QUANTIFICATION ANALYSIS

An ideal scientific model must not only produce accurate predictions but also reliably assess its own confidence. To comprehensively evaluate our framework's capability in this regard, we compare it against two strong baselines: a **GNN (MSE) w/ Dropout** model that uses the same MC Dropout mechanism as ours, and the gold-standard **Deep Ensembles**. We assess the epistemic uncertainty of each model's predictions on both in-distribution (ID) and out-of-distribution (OOD) samples across all three datasets, with the results presented in Figure 3.

The results exhibit a consistent trend across all three benchmarks, clearly demonstrating the superiority of our approach in learning a meaningful uncertainty representation. The **GNN (MSE) w/ Dropout** baseline shows only a marginal difference in uncertainty between ID and OOD samples in all tasks, indicating that standard supervised training combined with Dropout is insufficient to teach the model to distinguish between known and unknown physical regimes. The **Deep Ensembles** baseline, as expected, proves effective, generating higher uncertainty on OOD data than on ID data by leveraging the variance across multiple independently trained models. However, our method (`PhyRL`) delivers the most remarkable performance across all three datasets: it not only produces significantly higher uncertainty on OOD samples, but the **degree of separation** between its ID and OOD uncertainty distributions is the largest among all methods. This significant improvement is attributable to our $R_{\text{uncertainty}}$ reward, which directly incentivizes the generation of high uncertainty on OOD data during training. Consequently, our single model achieves and even surpasses the uncertainty quantification performance of Deep Ensembles at a fraction of the computational cost, effectively fostering the model's "self-awareness."

### 3.3 ABLATION STUDY

To thoroughly understand the individual contributions of the components within our Physics Reward Oracle, we conduct a comprehensive ablation study, with the results presented in Table 3. We systematically analyze the impact of the physical consistency reward ($R_c$) and the robustness reward ($R_r$) by selectively removing them from our full model. The evaluation metrics are specifically chosen to align with the primary objective of each reward component: NRMSE on OOD data to measure the impact of $R_c$ on generalization, and Robustness Error to assess the contribution of $R_r$ to stability.

The results reveal a clear "orthogonal effect" or "division of labor" in our reward design. When the physical consistency reward is removed (`PhyRL (w/o` $R_c$`)`), the model's error on the OOD test sets increases dramatically across all three datasets, regressing to a performance level comparable to

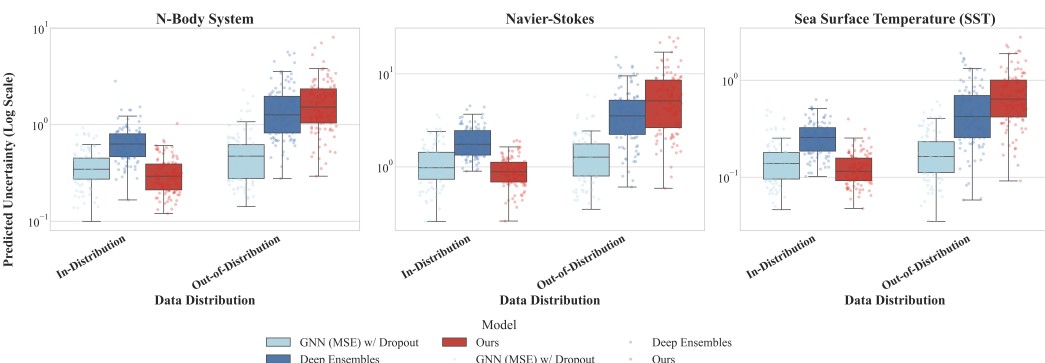

Figure 3: **Uncertainty Quantification Analysis.** The figure compares the distributions of predicted epistemic uncertainty for different models on in-distribution (ID) and out-of-distribution (OOD) data. Uncertainty is estimated as the variance from Monte-Carlo Dropout (or across the ensemble). Across all three datasets, our method (**Ours**) produces significantly higher uncertainty on OOD samples compared to ID samples, and the degree of separation between these distributions surpasses that of the baselines. This indicates that our framework effectively trains the model to recognize novel scenarios and express well-calibrated confidence, whereas the supervised model (GNN (MSE) w/ Dropout) is prone to overconfident predictions.

Table 3: **Ablation study of the reward components.** We analyze the contribution of each reward signal by selectively removing them from our full model. The metrics are chosen to reflect the primary goal of each component: NRMSE on OOD data for the consistency reward ($R_c$) and Robustness Error for the robustness reward ($R_r$). The results demonstrate a clear division of labor, where each component is critical for its targeted objective. Lower is better.

| | N-Body System | | | Navier-Stokes | | | Sea Surface Temperature (SST) | | |
|---|---|---|---|---|---|---|---|---|---|
| Method | NRMSE (ID) | NRMSE (OOD) | Robustness Err. | NRMSE (ID) | NRMSE (OOD) | Robustness Err. | NRMSE (ID) | NRMSE (OOD) | Robustness Err. |
| GNN (MSE) | 0.0213 | 0.0986 | 0.385 | 0.0701 | 0.2243 | 0.410 | 0.1002 | 0.2764 | 0.455 |
| PhyRL (w/o $R_c$) | 0.0221 | 0.0955 | 0.135 | 0.0655 | 0.1788 | 0.185 | 0.0945 | 0.2155 | 0.245 |
| PhyRL (w/o $R_r$) | 0.0205 | 0.0345 | 0.355 | 0.0633 | 0.0951 | 0.380 | 0.0928 | 0.1498 | 0.415 |
| PhyRL (**Full**) | **0.0195** | **0.0314** | **0.125** | **0.0611** | **0.0894** | **0.175** | **0.0913** | **0.1422** | **0.235** |

the PINN-style baseline, while its robustness error remains relatively stable. This provides strong evidence that $R_c$ is the core driver for enabling the model to learn generalizable physical laws and thus achieve superior OOD performance. Conversely, when the robustness reward is removed (**PhyRL (w/o $R_r$)**), the model still performs well on the OOD task but its robustness error escalates significantly, rendering it nearly as brittle as the purely supervised GNN (MSE) baseline. These findings not only validate the rationale behind our reward function design but also demonstrate that each component is both non-redundant and indispensable for achieving its targeted physical desideratum.

## 4 CONCLUSION

In this paper, we addressed the critical challenges of insufficient robustness and poor out-of-distribution (OOD) generalization in spatio-temporal Graph Neural Networks. We introduced `PhyRL`, a novel Physics-guided Reinforcement Learning framework. Our core contribution, the Physics Reward Oracle, successfully translates abstract physical principles, such as consistency, robustness, and uncertainty reliability, into quantitative reward signals. By leveraging reinforcement learning, `PhyRL` guides the model to move beyond mere data fitting and instead learn to internalize the intrinsic physical laws of the system. Extensive experiments across diverse and complex spatio-temporal tasks demonstrate that `PhyRL` significantly enhances the prediction accuracy, resilience to perturbations, and quality of uncertainty quantification for state-of-the-art GNN architectures, particularly in challenging OOD settings. This work offers a flexible and powerful new perspective on using verifiable reward-based RL paradigms to tackle fundamental problems in scientific computing, especially for improving the fidelity and generalization of complex models.

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

## A  THE USE OF LARGE LANGUAGE MODELS (LLMS)

LLMs were not involved in the research ideation or the writing of this paper.

## B  EVALUATION METRICS

In this section, we provide the detailed mathematical formulations for the evaluation metrics used in our experiments to assess model performance in terms of prediction accuracy, robustness, and uncertainty quality.

### B.1  NORMALIZED ROOT MEAN SQUARED ERROR (NRMSE)

**English Version**   The Normalized Root Mean Squared Error (NRMSE) is used to measure the prediction accuracy. It normalizes the standard Root Mean Squared Error (RMSE) by the standard deviation of the ground truth data, providing a scale-invariant error metric. A lower value indicates higher accuracy. The formula is:

$$\text{NRMSE} = \frac{\sqrt{\frac{1}{N} \sum_{i=1}^{N} (y_i - \hat{y}_i)^2}}{\text{std}(Y)} \tag{13}$$

where:

- $Y = \{y_1, y_2, \ldots, y_N\}$ is the set of ground truth values.
- $\hat{Y} = \{\hat{y}_1, \hat{y}_2, \ldots, \hat{y}_N\}$ is the set of predicted values.
- $N$ is the total number of data points (e.g., nodes $\times$ timesteps).
- $\text{std}(Y)$ is the standard deviation of the ground truth values $Y$.

For robustness evaluation, the "Robustness Error" is calculated as the NRMSE on predictions made from noise-perturbed initial states.

### B.2  EXPECTED CALIBRATION ERROR (ECE)

**English Version**   The Expected Calibration Error (ECE) is used to measure the quality of uncertainty quantification. It assesses how well the model's predicted confidence aligns with its actual accuracy. The confidence scores are partitioned into $M$ bins. The ECE is the weighted average of the absolute difference between the accuracy and confidence within each bin. A lower ECE indicates a better-calibrated model. The formula is:

$$\text{ECE} = \sum_{m=1}^{M} \frac{|B_m|}{N} \left| \text{acc}(B_m) - \text{conf}(B_m) \right| \tag{14}$$

where:

- $N$ is the total number of samples.
- $M$ is the number of confidence bins.
- $B_m$ is the set of predictions whose confidence falls into the $m$-th bin.
- $|B_m|$ is the number of predictions in bin $B_m$.
- $\text{acc}(B_m)$ is the accuracy of predictions in $B_m$. For regression tasks, this is often the fraction of ground truth values that fall within the predicted confidence interval.
- $\text{conf}(B_m)$ is the average confidence of predictions in $B_m$.

## C  RELATED WORK

**Spatio-temporal Graph Neural Networks for Dynamical Systems.**  Graph Neural Networks (GNNs) (Deng & Hooi, 2021; Boniol & Palpanas, 2020; Fan et al., 2019; Corso et al., 2024) have become a cornerstone for data-driven modeling of spatio-temporal dynamical systems (Ghadami & Epureanu, 2022; Müller, 2023; Yu et al., 2018; Mohan et al., 2020), owing to their intrinsic ability to operate on non-Euclidean data like meshes (Rame et al., 2022; Pfaff et al., 2020), particles, and sensor networks. State-of-the-art models have demonstrated remarkable success in learning complex physical interactions and accelerating simulations by orders of magnitude.  These architectures leverage message-passing mechanisms to explicitly model the spatial dependencies and evolve the system state over time. However, despite their impressive performance on in-distribution data, these models often function as powerful "curve-fitters." Their reliance on statistical correlations learned from the training set renders them brittle, leading to poor out-of-distribution (OOD) (Koh et al., 2021; Rame et al., 2022; Yang et al., 2022) generalization and a lack of robustness against perturbations, which are critical limitations this work aims to address.

**Integrating Physical Priors into Deep Learning.** To bridge the gap between data-driven models and physical reality, the field of Physics-Informed Machine Learning (PIML) has explored various strategies for embedding domain knowledge.  A dominant paradigm is Physics-Informed Neural Networks (PINNs) (Hao et al., 2022; Cho et al., 2023; Ren et al., 2022; Raissi et al., 2019; Cai et al., 2021), which incorporate physical laws, typically expressed as Partial Differential Equations (PDEs), as a soft constraint by adding the PDE residual to the loss function (Long et al., 2018; Takamoto et al., 2022; Lippe et al., 2023; Huang et al., 2023) Another powerful approach involves designing architectures with built-in symmetries, such as equivariant networks, which guarantee that the model's predictions respect fundamental principles like rotational or translational invariance. While these methods enhance generalization, they primarily treat physical laws as either a penalty to be minimized or a hard-coded architectural property. Our PhyRL framework proposes a distinct approach, reframing the problem from one of supervised learning to reinforcement learning. Instead of penalizing physically inconsistent outputs, we directly reward physically plausible and robust trajectories (Su et al., 2019), guiding the model's learning process towards a more fundamental understanding of the system's behavior (Liu et al., 2015; Hess et al., 2022).

**Reinforcement Learning in Scientific Applications.** Reinforcement Learning (RL) has traditionally excelled in decision-making and control tasks (Janner et al., 2021; Lange et al., 2012; Fernández & Veloso, 2006; Foerster et al., 2019; Xu et al., 2022), such as robotics and game playing. Recently, its application has expanded into scientific domains (Wang et al., 2023; Takamoto et al., 2022; Azizzadenesheli et al., 2024), including controlling plasma in fusion reactors, discovering novel molecules, and optimizing experimental designs.  In these contexts, RL agents typically interact with and control an external environment or simulation. Our work applies the RL paradigm in a novel manner: we do not control an external physical system, but rather the internal generative process of the predictive model itself. The GNN acts as a policy that generates future trajectories, the "environment" is the space of all possible physical evolutions, and the reward is directly quantified by our Physics Reward Oracle based on physical fidelity. This shifts the role of RL from an external controller to an internal training regularizer, shaping the model's behavior to intrinsically align with the laws of physics (Cai et al., 2021; Wang & Yu, 2021; Wang et al., 2024).

