# OpenReview forum: "Reinforcing Spatio-Temporal Graph Neural Networks with a Physics Reward Oracle"
_ICLR.cc/2026/Conference — Submitted to ICLR 2026_

### Official Review · Reviewer_v2Jd · 2025-10-16

**Soundness:** 2
**Presentation:** 3
**Contribution:** 3
**Rating:** 2
**Confidence:** 3

**Summary:**

The paper introduces PhyRL, a Physics-Guided Reinforcement Learning framework that improves the robustness and out-of-distribution generalization of spatio-temporal Graph Neural Networks. A Physics Reward Oracle provides rewards based on physical consistency, robustness, and uncertainty reliability, guiding the model via reinforcement learning rather than standard loss minimization. Experiments on N-body, Navier–Stokes, and sea surface temperature tasks show that PhyRL significantly enhances accuracy, stability, and uncertainty calibration, offering a new reward-driven approach to physics-informed modeling.

**Strengths:**

- The paper presents a novel perspective by reframing physics-informed learning as a reinforcement learning problem, where physical laws are not imposed as soft constraints but embedded as verifiable reward signals.
- The work addresses the limitation of current spatio-temporal GNNs—poor OOD generalization—and introduces a broadly applicable framework for improving the physical fidelity and reliability of data-driven dynamical models.
- The paper is well-organized and clearly written, with intuitive figures (e.g., Fig. 1) that effectively illustrate the dual-phase framework.

**Weaknesses:**

- Key implementation and training information are missing. The GNN architecture, baseline configurations, and RL training details (e.g., learning rate, steps, reward curves) are not provided. Important hyperparameters ($w_c, w_r, w_u, \lambda_c, \lambda_r, \lambda_u$) and their sensitivity are also unreported, making the results less convincible.
- The framework’s computational cost is not discussed. Reinforcement learning loops can be significantly more expensive than supervised training, and the paper provides no analysis of training efficiency or scalability.
- Missing related work[1], which also integrates reinforcement learning into GNN-based physical simulations through adaptive remeshing.
- Although the paper claims to release code, the provided anonymous link only contains a README without actual implementation.

[1] Learning Controllable Adaptive Simulation for Multi-resolution Physics.
Tailin Wu, Takashi Maruyama, Qingqing Zhao, Gordon Wetzstein, Jure Leskovec. ICLR 2023

**Questions:**

1. In line 192, you mention computing the consistency loss by integrating the squared PDE residual over the spatio-temporal domain, with differential operators approximated on the graph. Could you provide the explicit computational formula or an example of how this residual is implemented on discrete graph structures?
2. The reward terms (e.g., R = $\exp(-\lambda v)$ ) all adopt an exponential transformation. What is the rationale behind this specific functional form? Have you compared it with other mappings such as linear or logistic functions?
3. In line 203, when defining the robustness reward, how is the perturbation noise intensity ($\sigma$) determined? Is it fixed across datasets or tuned per task?
4. The uncertainty reward relies on Monte Carlo Dropout, which requires dropout layers. How would this be applied to architectures like Transformers or Graph Neural Operators that lack dropout modules?
5. During Physics-Guided Tree Search, are the multiple candidate trajectories generated through MC Dropout sampling? If so, how many samples are drawn at each step, and how sensitive is the search result to this number?
6. There are typos in Figure 2 (“\textbf{Ours}”), which should be corrected in the final version.

---

### Official Review · Reviewer_eRwP · 2025-10-27

**Soundness:** 2
**Presentation:** 2
**Contribution:** 2
**Rating:** 2
**Confidence:** 5

**Summary:**

As stated in my Official Comment, this submission raises serious concerns about compliance with ICLR 2026 policies (**i.e., LLM Usage Statements**). In light of these issues, I have recommended a **desk rejection**. The remarks below are offered only for completeness, as the paper in its current form should not proceed to full review.

**Strengths:**

N/A

**Weaknesses:**

N/A

**Questions:**

N/A

---

### Official Review · Reviewer_Q7ua · 2025-10-27

**Soundness:** 2
**Presentation:** 2
**Contribution:** 2
**Rating:** 4
**Confidence:** 3

**Summary:**

The paper examined the using reinforcement learning to guide the learning of a spatio-temporal GNN to generate trajectories to learn the physical laws of the system. The physical reward oracle consists of 3 different terms:
- a consistency term which sees how the trajectories follow the physical law of the system (usually in the form of a PDE) which is important for the model to be able to generalize to OOD states
- a robustness term which sees how the model is affected by different values of input perturbations
- a uncertainty term uses Monte Carlo dropout in the GNN to assign a high uncertainty value to OOD trajectories and low uncertainty values to ID trajectories.

Experiments involve comparing the proposed model to other GNN models which are trained with supervised learning (rather than RL) as well as other models which incorporate physical laws via a soft constraint. Ablations are also performed to examine the effect of the different terms in the loss function.

**Strengths:**

- The paper focues on a novel area of interest which is using graph neural network with reinforcement learning to learn the behavior of physical systems, and combines the different components in a original way. Additionally, the work tackles the problem of OOD generalization in a unique way.
- The paper is well-written and is clear, particulaly the explanation of the different temrs in the physical reward oracle and how it affects the learning of the system.
- Clear experiments done on several different datasets with several different baselines. The model is able to outperform the other models on both the ID and OOD trajectories, highlighting that the model is better able to generalize to OOD samples.
- Clear illustrative figures and appropriate ablations to get a better understanding of the different components of the physical reward oracle.

**Weaknesses:**

- It was difficult to understand how the inference process works synergetically with the training of the model.
- There was not much discussion on the selection of the different hyperparameters (such as the weighing terms of the reward function) and how these hyperparameters would affect the model.
- The comparison for the uncertainty estimates involved deep ensembles and drop-out I believe, these models were not given OOD samples during training and therefore it may be unfair that the authors model is compared with these when the authors model is given OOD samples during training,
- This macro-action regime is unconvincing, since you define the action space in the same space as the state space. It seems like what you are doing is learning a model/transition function (like in model-based RL) than learning a policy. Feels more natural to say that you are learning the function M(s_{t+1}|s_t,a_t) (here in the spacial case where no action is taken, since no exogenous variable affects the dynamics) than a policy P(a_t|s_t).

In summary, it is not clear to me why you frame this as a policy learning problem as opposed to a (world) model learning problem. It seems like this was done to take advantage of developments in the RL community like PPO. But it is not clear that the exact benefit of this setup is. I would suggest clearly justifying why this is inevitably a RL problem.


### Minor issues (no effect on the rating):
- The code is absent.
- Small text error in one of the plots related to using \textbf

**Questions:**

1) Could you further describe how the physics based tree search works and how this can aid the training?
2) Was there any choosing of the selection of the hyperparameters, or how robust is the model to the choice of hyperparameters (for the weighting terms in the physical reward oracle)?
3) As mentioned in weaknesses, I don't fully understand the action-states in the RL setting. As I understand, you are calling an action the result of applying the network for future trajectories. Furthermore, the presence of a Physics oracle makes this very similar to supervised learning. Please, can you explicitly clarify what makes this method better than supervised learning?
4) Please explain why treat the sequence of future states in Eq (1) as an action and not as a state ?
5) The residual loss function in Eq (4) is very similar to the classical PINN, which, arguably, is more straightforward than the proposed approach, to achieve the same goal. It was good practice by the authors to compare their method to the PINN, but what steps were taken to ensure a fair comparison (e.g., what terms were used in the PINN's multi-objective loss function) ?
6) Finally, the physics reward is tied to a specific equation, which means if any underlying physical parameter is changed (e.g. $D$ in Eq (4)), the model would need to be retrained from scratch, just like with PINNs. This is a major limitation, and has been addressed in recent papers [1,2,3], etc. I have two sub questions here:
     - Is the cost (re)training light enough to address this limitation? (Essentially, what is the computational cost of this method: memory, wall clock time, etc.?)
     - Are there any preliminary ideas to tackle this in future work ?



### References:
- [ 1 ] Kirchmeyer et al., Generalizing to New Physical Systems via Context-Informed Dynamics Model, ICML 2022
- [ 2 ] Nzoyem et al., Neural Context Flows for Meta-Learning of Dynamical Systems, ICLR 2025
- [ 3 ] Le Boudec et al., Learning a Neural Solver for Parametric PDE to Enhance Physics-Informed Methods, ICLR 2025

**Details Of Ethics Concerns:**

I agree with Reviewer eRwP that there should be further investigations into the use of LLMs for writing Appendix B of the paper.

---

### Official Review · Reviewer_taDi · 2025-10-29

**Soundness:** 2
**Presentation:** 3
**Contribution:** 2
**Rating:** 0
**Confidence:** 4

**Summary:**

The paper introduces a new training approach targeted at improving out-of-distribution performance of GNNs. It is based on a reinforcement learning paradigm and defining a reward model that incorporates physical priors or symmetries as well as robustness to slight perturbations. The approach is evaluated on synthetic settings of N-body problem and Navier-Stokes PDE as well as real-data of sea-surface temperature prediction.

**Strengths:**

- Tackles the important problem of out-of-distribution generalization, which is mostly open for dynamical systems
- Clearly written paper
- Applies reinforcement learning to forecasting/ prediction tasks which is novel in scientific computing

**Weaknesses:**

- Focuses on settings with known physical laws, hence is limited by design
- Lacks a principled justification for the improved performance compared to approaches that incorporate physical priors directly
- Might be data-intensive

**Questions:**

1/ What's the mount of data used ?

2/ Could you include more details on severity of distribution-shift (e.g. viscosity coefficients for Navier-Stokes)

3/ How does the approach scale in terms of computational cost ?

**Details Of Ethics Concerns:**

While the authors assert that “LLMs were not involved in the research ideation or the writing of this paper,” evidence within the manuscript suggests otherwise. Specifically, Appendix Sections B.1 and B.2 contain headings labeled “English Version,” which are unrelated to the description of evaluation metrics. This terminology implies that parts of the text were initially composed in another language and later translated—most likely with the assistance of an LLM—leaving behind traces of that process in the final document.

In light of these issues, I recommend rejection in accordance with ICLR policy.

---

### Meta-Review · Area_Chair_H9mo · 2025-12-15

**Summary:**

This paper proposes a  Physics-Guided Reinforcement Learning framework (PhyRL) that improves the robustness and out-of-distribution generalization of spatio-temporal Graph Neural Networks.  PhyRL introduce a  "Physics Reward Oracle" to provide verifiable, quantitative rewards for multiple candidate future trajectories generated by the model.

**Reviewer Concerns:**

The concerns would still outstanding after rebuttal:
- ethics concerns raised by Reviewer `taDi ` and `eRwP`.
- unconvincing macro-action regime.
- the justification of the exact benefit of policy learning setting, rather than model learning setting.

**Reviewer Scores:**

All reviewers tend to reject this paper. I think it is hard to raise it to a positive score, even after the rebuttal.

---

### Decision · Program_Chairs · 2026-01-26

Reject